# Comparison of the Gut Microbial Communities of Domestic and Wild Mallards (*Anas platyrhynchos*) Based on High-Throughput Sequencing Technology

**DOI:** 10.3390/ani13182956

**Published:** 2023-09-18

**Authors:** Yaoyin He, Minghui Zhang, Chuanyin Dai, Lijiang Yu

**Affiliations:** 1Animal Science and Technology College, Guangxi University, Nanning 530004, China; hyy2118391007@163.com (Y.H.); 17356533778@163.com (M.Z.); 2Key Laboratory of Ecology of Rare and Endangered Species and Environmental Protection (Guangxi Normal University), Ministry of Education, Guilin 541006, China; daicy527@163.com

**Keywords:** bird, gut microbiota, mallard (*Anas platyrhynchos*), 16S rRNA gene, potential pathogenic genera

## Abstract

**Simple Summary:**

Gut microbes play an important role in maintaining normal physiological functions of the host and are influenced by a number of listed biotic and abiotic factors, including host species, habitat environment, diet, and others. This study compared the composition and diversity of the gut microbial community of mallards (*Anas platyrhynchos*) in two different habitat environments. The results showed that the gut microbial community significantly differed between domestic and wild mallards. In addition, we found that the mean relative abundance of five potential pathogenic genera in the feces of domestic mallards was higher than that of wild mallards. This study provides basic information for the conservation of wild populations and the prevention and control of diseases in domestic mallards.

**Abstract:**

Mallards (*Anas platyrhynchos*) are currently one of the most popular species in rare bird breeding in several southern provinces of China, but there have been no studies comparing the gut microbial communities of domestic and wild mallards. In this study, 16S rRNA gene high-throughput sequencing technology was used to compare the composition and diversity of gut microbial communities in domestic and wild mallards. Alpha diversity analysis showed significant differences in gut microbial communities between the two groups of mallards, and the diversity and richness of gut microbial communities were significantly higher in wild mallards than in domestic mallards. Beta diversity analysis showed that the two groups of stool samples were mostly separated on the principal coordinate analysis (PCoA) plot. In domestic mallards, Firmicutes (68.0% ± 26.5%) was the most abundant bacterial phylum, followed by Proteobacteria (24.5% ± 22.9%), Bacteroidetes (3.1% ± 3.2%), Fusobacteria (2.2% ± 5.9%), and Actinobacteria (1.1% ± 1.8%). The dominant bacterial phyla in wild mallards were Firmicutes (79.0% ± 10.2%), Proteobacteria (12.9% ± 9.5%), Fusobacteria (3.4% ± 2.5%), and Bacteroidetes (2.8% ± 2.4%). At the genus level, a total of 10 dominant genera (*Streptococcus*, *Enterococcus*, *Clostridium*, *Lactobacillus*, *Soilbacillus*, *Bacillus*, *Acinetobacter*, *Comamonas*, *Shigella*, and *Cetobacterium*) with an average relative abundance greater than 1% were detected in the fecal samples of both groups. The average relative abundance of five potential pathogenic genera (*Streptococcus*, *Enterococcus*, *Acinetobacter*, *Comamonas*, and *Shigella*) was higher in domestic mallards than in wild mallards. The enrichment of pathogenic bacteria in the intestinal tract of domestic mallards should be of sufficient concern.

## 1. Introduction

The collection of all microbial species and their associated genetic material in the host gut is called the gut microbiome [1]. The gut microbiota is considered the “second genome” of the animal and is intricately linked to the host. An increasing number of studies have shown that gut microbes play important physiological functions, such as nutrient absorption [2], energy supply and storage [3], development and maintenance of the intestinal mucosal barrier [4], establishment of the immune system, and resistance to pathogens [5]. Accordingly, when dysbiosis occurs and the normal balance between the gut microbiota and the host is disrupted, the host is more susceptible to a variety of diseases [6], such as obesity [7], cancer [8], diabetes [9], inflammatory bowel diseases [10,11], and other neurological diseases [12]. Therefore, the study of gut microbes is of physiological importance.

Birds are an important member of the biodiversity family and an important indicator species in ecosystems. More than 10,000 species of birds have been identified worldwide [13], more than twice as many as mammals. Research on wild birds’ gut microbes lags far behind that of mammals because of factors such as the difficulty of sampling wild birds under natural conditions and the difficulty of preserving samples. Birds have complex and unique foraging strategies, physiological characteristics, and phylogenetic relationships [14]. It has been suggested that the gut microbial community of birds may be influenced by a range of biotic and abiotic factors, including host species [15,16], habitat environment [17,18,19,20], diet [21,22,23,24], season [25], and sex [26]. Among these, habitat environment and diet are considered to be the main factors shaping the diversity of gut microbial communities in birds. For example, gut microbial composition and diversity tend to be more similar between the same species in the same habitat than between species that are distantly related [27,28]. Significant differences in gut microbial communities may also exist between the same species in different geographical locations [29].

With the rapid development of high-throughput sequencing technologies, coupled with efficient bioinformatics analysis tools, it has been possible to conduct in-depth research on the vertebrate microbiome. An earlier report on the application of high-throughput sequencing technology to the intestinal flora of birds was seen in 2013 in a study on the diversity of the cecum flora of emus (*Dromaius novaehollandiae*) [30]. Current research on gut microbes in captive and wild birds is focused on a few species of economic and conservation value. For example, Wang et al. showed significant differences in microbial communities between wild and farmed swan geese (*Anser cygnoides*) in a study of their gut microbes [31]. In addition, Wang et al. also compared the intestinal microbial communities of bar-headed goose (*A. indicus*) in three different rearing modes: artificial breeding, semi-artificial breeding, and wild and found that the highest diversity and richness of gut microbial communities were found in the semi-artificial breeding group, followed by the wild group, and, finally the artificial breeding group [32]. Jiang et al. studied the gut microbial composition of wild and captive Chinese monals (*Lophophorus ihuysii*), showing that the alpha diversity of the gut microbial community was significantly higher in the wild group than in the captive group and that the core bacterial groups of the two groups differed significantly at the level of phylum, class, order, and family [33]. Xie et al. conducted research on three groups of red-crowned cranes (*Grus japonensis*) and showed that captive cranes had the greatest gut microbial alpha diversity, while wild cranes had the lowest gut microbial alpha diversity [34]. In summary, different habitat environments affect the composition of bird gut microbial communities and produce significant differences between wild and captive individuals. In response, we propose two scientific questions in this article. First, what is the composition of the gut microbial community in domestic and wild mallards? Second, are there significant differences in the gut microbial communities of different habitat environments for the same species?

The wild mallard is one of the ancestors of domestic ducks in China [35], has a long history of domestication and breeding, and is one of the most popular species in rare wildfowl farming in China. Domestic mallards have the advantages of strong disease resistance, wide adaptability, a diversified diet, high feed remuneration, and a short feeding cycle. Tian et al. investigated the structural and functional characteristics of the intestinal microflora of Shaoxing ducks (*A. platyrhynchos*) in a high-temperature environment [36]. Their results showed that Firmicutes was the most abundant dominant phylum in the intestinal tract of Shaoxing ducks, and its relative abundance increased with increasing temperature. However, no comparisons of the intestinal flora of wild and farmed mallards have been carried out. As far as we know, this article is the first study of the gut microbial community of wild mallards and the first to use high-throughput sequencing of the 16S rRNA gene (i) to compare the composition and diversity of gut microbial communities in domestic and wild mallards; (ii) to analyze potential drivers influencing the composition of gut microbial communities; and (iii) to identify potential pathogenic genera threatening mallard health. The results of this study provide insight into the composition of the gut microbial community of mallards and provide a theoretical basis for their conservation, especially for disease prevention and control.

## 2. Materials and Methods

### 2.1. Wild Mallards Fecal Sample Collection

Currently, cloacal, fecal, and other nonlethal sampling has become the main source of samples for studying the gut microbial communities of birds [1,37,38]. This noninvasive method is easy to perform, has a low impact on individual birds, and has been widely used in the study of gut microbiota in wild birds. Multiple lines of evidence suggest that fecal samples provide a truer picture of the gut microbial community than cloacal samples [39], making fecal samples an alternative to studying the gastrointestinal microbiota [40].

Eight wild mallard fecal samples were collected from Caohai National Nature Reserve (26.84921854° N, 104.28497294° E), Guizhou Province, China, in March 2022. First, binoculars were used to observe the range of the mallards, waiting for the ducks to fly away to collect fresh droppings immediately. To avoid collecting samples from the same mallards, samples were taken at a minimum distance of 5 m apart, and only the middle part of the feces was collected. The stools were collected in sterile 15 mL centrifuge tubes filled with ethanol [41] and transported to the laboratory for freezing and storage at −80 °C.

### 2.2. Domestic Mallards Fecal Sample Collection

Twelve fecal samples of domestic mallards were collected from June to August 2022 at Guangxi University. Sampling and preservation methods were as above. In addition, we considered that the domestic mallard rearing environment is relatively stable and its influence by seasonal factors is small; therefore, we excluded the interference of seasonal factors on the mallard intestinal microbial community in this study.

### 2.3. Fecal DNA Extraction and PCR Amplification

The genomic DNA of the samples was extracted using the CTAB method, and the purity and concentration of DNA were checked by 1% agarose gel electrophoresis. An appropriate amount of sample was taken in a centrifuge tube and the sample diluted to 1 ng/µL with sterile water. PCR amplification of the V3-V4 variable region was performed using 341F (5′-CCTAYGGGRBGCASCAG-3′) and 806R (5′-GGACTACNNGGGTATCTAAT-3′) primers. PCR reaction system (30 µL): Phusion Master Mix (2×) 15 µL; forward primer (1 µM/µL) 1 µL; reverse primer (1 µM/µL) 1 µL; gDNA (1 ng/µL) 10 µL; H_2_O 2 µL. Reaction procedure: 98 °C pre-denaturation for 1min; 30 cycles including (98 °C, 10 s; 50 °C, 30 s; 72 °C, 30 s); 72 °C, 5 min.

### 2.4. Mixing and Purification of PCR Products

PCR products were mixed in aliquots according to the concentration of the PCR product and, after thorough mixing, the PCR products were purified by agarose gel electrophoresis using 1 × TAE at 2% concentration, and the target bands were recovered using the Universal DNA Purification and Recovery Kit (TianGen, Beijing, China).

### 2.5. Libraries Generated and IIIumina NovaSeq Sequencing

Sequencing libraries were generated using the NEB Next^®^ Ultra DNA Library Prep Kit (Illumina, San Diego, CA, USA) following the manufacturer’s recommendations, and index codes were added. The library quality was assessed on the Agilent 5400 (Agilent Technologies Co., Ltd., Santa Clara, CA, USA). Finally, the libraries were sequenced on the Illumina NovaSeq platform to generate 250 bp paired end reads.

### 2.6. Data Analysis

The analysis was conducted by referring to the tutorial “Atacama soil microbiome tutorial” (https://docs.qiime2.org/2019.1/, accessed on 5 August 2022) in the QIIME2 documentation. The QIIME Tools import plugin was used to import the original sequence fastq file into a file format that can be processed by QIIME2. Demultiplexed sequences from each sample were quality-filtered and trimmed, de-noised, and merged, and then the chimeric sequences were identified and removed using the QIIME2 dada2 plugin (https://docs.qiime2.org/2019.1/, accessed on 5 August 2022) to obtain the feature table of an amplicon sequence variant (ASV). Next, the QIIME2 feature-classifier plugin was applied to match the representative sequences of ASVs to the pre-set GREENGENES database with 99% similarity of version 13_8 (the database was pruned to the region of V3-V4 based on the 341F/806R primer pairs), and the taxonomic information table of the species was obtained. Any contaminating mitochondrial and chloroplast sequences were filtered using the QIIME2 feature table plugin. The diversity matrix was then calculated using the QIIME2 core-diversity plugin. Characteristic sequence level alpha diversity indices (including the Shannon index and the Chao1 index) were used to assess the degree of diversity of the samples themselves. Beta diversity indices, including weighted UniFrac and unweighted UniFrac indices, were used to assess the structural variability of microbial communities between samples and were subsequently presented using PCoA plots. The Wilcoxon test and Bray-Curtis dissimilarity matrix were used to identify bacteria that differed in abundance between groups and samples. A *p* value < 0.05 indicates significant differences, and *R*^2^ values indicate the extent to which differences are explained. Gut bacterial taxa that differed significantly between the two groups of mallards were identified using linear discriminant analysis (LDA) effect size (LEfSe), which identifies biomarkers using a nonparametric Kruskal-Wallis rank-sum test at the default settings (*p* value of 0.05, effect size threshold of 3.0). The functions of the bacteria with significant abundance (mean relative abundance ± SD) were predicted using the Kyoto Encyclopedia of Genes and Genomes (KEGG) database. Unless otherwise noted, the parameters applied in the above analysis are the default settings.

## 3. Results

### 3.1. 16S rRNA Gene Data

Using the dada2 plugin of QIIME2 to quality control, de-noising, merge, and de-chimerism all raw sequences of all samples, we obtained 1,024,791 high-quality sequences (Appendix A), with an average of 51,239 ± 12,487 sequences per sample (mean ± SD). A total of 7555 ASVs were identified based on the 99% sequence similarity threshold, and these ASVs were classified into 38 phyla, 101 classes, 168 orders, 236 families, and 409 genera (Appendix A). The rarefaction curve indicated that new species could not be detected by continuing to increase the sequencing depth (Figure 1A). As shown in Figure 1B, the Shannon diversity curve gradually reached a plateau as the sequencing depth increased, which indicates that the sequencing depth is sufficient to reflect the diversity of the samples and the data are suitable for subsequent analysis.

### 3.2. Alpha Diversity and Beta Diversity Analyses

We analyzed the diversity and richness of microbial communities in our samples by Shannon index (Figure 2A) and Chao1 index (Figure 2B). The results showed that there were significant differences between the diversity and richness of the gut microbial communities of the two groups of mallards (Shannon index: *p* = 0.02; Chao1 index: *p* = 0.03), and the diversity and richness of the gut microbial communities of the WM group were significantly higher than those of the DM group (Figure 2).

The composition of microbial communities was compared between samples by principal coordinate analysis (PCoA). According to the weighted (*p* = 0.002; *R*^2^ = 0.385) and unweighted UniFrac (*p* = 0.242; *R*^2^ = 0.036) distances, the two groups of samples were mostly separated (Figure 3). This suggests that gut microbial communities differ between the same species.

### 3.3. Comparison of the Intestinal Microflora of Two Groups of Mallards at the Phylum and Genus Levels

A total of 38 bacterial phyla were identified in the 20 stool samples, and Figure 4A shows the average relative abundance of bacterial phyla in the top 10 in both groups. In the wild mallard (WM) group, the total sequences were identified as four major phyla; Firmicutes were absolutely dominant with an average relative abundance of 79.0% ± 10.2%, followed by Proteobacteria (12.9% ± 9.5%), Fusobacteria (3.4% ± 2.5%), and Bacteroidetes (2.8% ± 2.4%). In contrast, in the domestic mallard (DM) group, Firmicutes was also the most abundant, with an average relative abundance of 68.0% ± 26.5%, followed by Proteobacteria (24.5% ± 22.9%), Bacteroidetes (3.1% ± 3.2%), Fusobacteria (2.2% ± 5.9%), and Actinobacteria (1.1% ± 1.8%). Table 1 lists the bacterial phyla with an average relative abundance greater than 1% in both groups. The average relative abundance of the Firmicutes and Fusobacteria in the WM group was higher than that in the DM group, but the average relative abundance of the Proteobacteria, Bacteroidetes, and Actinobacteria in the WM group was lower than that in the DM group.

At the genus level, a total of 409 bacterial genera were identified from the 20 sample sequences. The top 10 bacterial genera in terms of average relative abundance are listed in Figure 4B, and sequences that could not be classified as known genera are indicated as “unclassified”. The dominant bacterial genera with an average relative abundance greater than 1% are listed in Table 2. In the DM group, the average relative abundance of *Streptococcus*, *Enterococcus*, *Acinetobacter*, *Comamonas*, and *Shigella* is higher than that in the WM group. In the WM group, the average relative abundance of *Clostridium*, *Lactobacillus*, *Cetobacterium*, *Solibacillus*, and *Bacillus* is higher than that in the DM group.

Significant differences were analyzed for the intestinal flora of mallards in both groups (Figure 5A). LEfSe analysis showed that, at the phylum level (Figure 5B), the biomarkers that differed significantly between the two groups of mallards were Fusobacteria and Actinobacteria (LDA > 3.0, *p* < 0.05). At the genus level (Figure 5B), the biomarkers that differed significantly between the two groups of mallards were *Cetobacterium*, *Lactobacillus*, *Solibacillus*, *Clostridium*, *Sporosarcina*, *Allobaculum*, *Bacillus*, *Sphingomonas*, *Paenibacillus*, *Halomonas*, *Flavobacterium*, *Rheinheimera*, *Desulfomonile*, *Bacillus*, *Shewanella*, *Mycoplasma*, and *Exiguobacterium* (LDA > 3.0, *p* < 0.05).

### 3.4. Prediction of Gut Microbiome Function

We predicted gut microbial function in domestic and wild mallards by PICRUSt2 analysis and annotated 47 functional pathways at KEGG level 2. Among the gut microorganisms of domestic mallards, carbohydrate metabolism (11.5% ± 0.9%), amino acid metabolism (10.0% ± 0.3%), metabolism of cofactors and vitamins (9.1% ± 0.6%), metabolism of other amino acids (8.0 ± 0.4%), global and overview maps (5.6% ± 0.2%), replication and repair (5.4% ± 0.7%), lipid metabolism (5.3% ± 0.4%), and biosynthesis of other secondary metabolites (5.1% ± 0.4%) of the KEGG pathway were very abundant. Similarly, the gut microorganisms of wild mallards had high abundances in amino acid metabolism (10.5% ± 0.4%), carbohydrate metabolism (10.3% ± 0.5%), metabolism of cofactors and vitamins (9.8% ± 0.2%), metabolism of other amino acids (8.0% ± 0.3%), lipid metabolism (6.2% ± 0.6%), global and overview maps (5.5% ± 0.1%), and replication and repair (5.2% ± 0.4%). Figure 6 shows the top 10 level 2 pathways in the DM and WM groups in terms of predicted abundance of gut microbial function.

## 4. Discussion

The composition and diversity of the bacterial community are considered important parts of the gut microbiome, and gut microbes, as a dynamic ecosystem, are easily influenced by multiple factors. In most of the available studies, diet is considered to be the underlying cause of the differences in the gut microbial community. Birds in different locations feed on different foods, and microorganisms ingested with the food may be one of the main pathways for microbial colonization of the gastrointestinal tract of birds [42]. In addition, the habitat environment sometimes even outweighs genetic factors in the formation of gut microbial communities in birds [43]. Birds are exposed to different microorganisms through the environmental conditions of their habitat (including diet, water, soil, nesting, and social activities), which are potential sources of microorganisms in the gastrointestinal tract of birds.

In this paper, we studied the gut microbial communities of domestic and wild mallards based on 16S rRNA gene high-throughput sequencing technology. Alpha diversity analysis confirmed significant differences in diversity and richness of the gut microbiota between domestic and wild mallards. This difference may be related to the fact that domestic and wild mallards live in different habitats and feed on different diets. This result is consistent with the study by Jiang et al. [33] on the gut microbiota of wild and captive Chinese monals. PCoA revealed that the two groups of mallards’ gut microbial communities were mostly separated. The higher specificity of gut microbes in the two groups of mallards indicated that there was no correlation between close evolutionary relationships and the composition of gut microbial communities. From this, we hypothesized that habitat environment and diet have a greater influence on the gut microbial community of mallards than genetic factors.

The gut microbial community composition of the two groups of mallards showed that Firmicutes, Proteobacteria, Bacteroidetes, Fusobacteria, and Actinobacteria were the dominant phyla, with an average relative abundance of more than 1%. These dominant phyla are similar to studies of gut microbiota in other wild birds, such as the Shaoxing duck (*A. platyrhynchos*) [36], bar-headed goose [44], whooper swan (*Cygnus cygnus*) [45], black-necked crane (*Grus nigricollis*) [46], hoatzin (*Opisthocomus hoazin*) [47], and turkey (*Meleagris gallopavo*) [48]. Firmicutes are the most prevalent and common bacterial phylum among all vertebrates. In mice and humans, Firmicutes have been shown to positively correlate with the ability to obtain energy and nutrient absorption from food [49,50]. Currently, we have not found any research on the function of Firmicutes in wild birds, but studies in domestic chickens have found a positive correlation between the abundance of Firmicutes and body weight gain and immune function. We speculate that Firmicutes may have similar roles in mammals and birds [51,52].

Proteobacteria were the second-most abundant bacterial phylum in both stool samples. The function of Proteobacteria in the gut of wild birds is unknown and further studies are needed to confirm it. According to previous studies, Proteobacteria have multiple physiological functions, are able to utilize a large amount of carbon sources and play an important role in the energy accumulation of the host [53,54,55]. The phylum Proteobacteria includes five major groups (α, β, γ, δ, and ε) that vary greatly in their occurrence and function inside and outside the gastrointestinal tract, with α proteobacteria being involved in the degradation of acidic herbicides [56], suggesting a possible detoxification role in the gastrointestinal tract of mallards.

Actinobacteria are the dominant bacterial phylum of the domestic mallard group. Turnbaugh et al. [57] showed that increased abundance of Actinobacteria was strongly associated with obesity and that 75% of obesity-enriched genes (involved in carbohydrate, lipid, and amino acid metabolism) were from Actinobacteria. In addition, the abundance of actinomycetes in stool samples from young children was reported to be positively correlated with the intake of barley dietary fiber [58]. Therefore, the physiological functions of actinomycetes in the gastrointestinal tract of mallards may be similar. However, more in-depth studies are needed to elucidate the role of specific members of the Actinobacteria phylum in the nutrition and health of mallards.

In this study, there were 10 bacterial genera with an average relative abundance greater than 1% (Table 2), which were distributed in three bacterial phyla. Among them, there are six genera (*Streptococcus*, *Enterococcus*, *Clostridium*, *Lactobacillus*, *Solibacillus*, and *Bacillus*) belonging to Firmicutes, Proteobacteria has three genera (*Acinetobacter*, *Comamonas*, and *Shigella*), and only one genus (*Cetobacterium*) belongs to Fusobacteria. Five potential pathogenic genera (*Streptococcus*, *Enterococcus*, *Acinetobacter*, *Comamonas*, and *Shigella*) were identified in the DM group with a higher mean relative abundance than in the WM group. *Lactobacillus* and *Bacillus* are two potential probiotics in wild mallards, and *Lactobacillus* can enhance host digestion and inhibit the development of certain diseases [59]. Antibiotics produced by *Bacillus* have a broad antibacterial spectrum, can bind lipopolysaccharides, and neutralize endotoxin. Probiotics prepared with *Bacillus* can regulate the balance of animal intestinal flora, enhance the immune function of animals, and promote the development of the animal intestinal tract.

A previous study found a negative correlation between the diversity of the gut microbial community and pathogenic bacteria in the gut [60], which is similar to the findings of this paper. This suggests that domestic mallards harbor more pathogenic bacteria in their intestines than wild mallards. *Streptococcosis* is a general term for a variety of animal and human infections caused primarily by *β*-hemolytic streptococci, which can cause many serious diseases such as meningitis and toxic shock in humans [61]. *Streptococcus* is highly susceptible, and a variety of poultry (ducks, geese, and chickens) can be infected; the main clinical manifestation is acute septicemia, and some are chronic infections. *Enterococcus* are opportunistic pathogens prevalent in the gastrointestinal tract of humans and a variety of animals (mammals, reptiles, birds, and some invertebrates) [62] and can cause serious infections such as endocarditis, septicemia, and urinary tract infections [63]. They have become the third most prevalent nosocomial pathogen in the world [64]. *Acinetobacter* is receiving increasing attention because of its strong resistance to antibacterial drugs. The environment, soil, and animals are the natural habitats of *Acinetobacter*, which infects humans by contaminating food and water. *Acinetobacter* has been isolated from various animal sources, including birds [65], poultry (chicken and turkey), cattle, pigs [66], fish [67], etc. *Acinetobacter* is associated with diseases such as septicemia, pulmonary infections, meningitis, and diarrhea in humans and animals, with a mortality rate of about 20–60% [68]. *Comamonas* is a Gram-negative pathogenic bacterium that causes steroid hormone degradation [69,70], is primarily associated with bacteremia [71], and occasionally causes low-virulence disease in humans and animals [72]. *Shigella* is a common and potentially pathogenic enteric pathogen that can cause bacterial food poisoning, typhoid fever, and uremia but is also present in small amounts in the feces of healthy individuals [73]. *Comamonas*, *Acinetobacter*, and *Shigella* all belong to Proteobacteria and it has been shown that an increase in the relative abundance of Proteobacteria can lead to the development of intestinal diseases and reduced production performance in chickens [74,75]. Though a certain percentage of pathogens are detected at the genus level, not all species within these pathogenic genera are pathogenic. We cannot arbitrarily state that these pathogenic genera are pathogenic to domestic mallards, as they are also frequently isolated in healthy birds [76]. Most of the potentially opportunistic pathogenic bacteria in the gut microbiota act more as commensals and may in fact be beneficial to the host [77]. However, if the environmental conditions in the gut change or if the population of these potentially opportunistic pathogens increases to pathogenic levels [78,79], the gut microbial ecosystem can be disrupted, endangering host health [77]. Thus, it is necessary to monitor the gut microbiota of domestic mallards on a regular basis to prevent the occurrence and spread of certain diseases.

## 5. Conclusions

It is important to study wild populations of species associated with poultry livestock to assess the potential health risks associated with these wild populations and to complete our ecological understanding of some pathogenic microorganisms. In this study, we used high-throughput sequencing of the 16S rRNA gene to analyze for the first time the intestinal microflora of domestic and wild mallards. Significant differences were found between the two groups of mallard gut microbial communities in terms of abundance and diversity, which may be closely related to the different diets and different living environments of domestic and wild mallards. PCoA showed that gut microbial communities differed between the same species. From this, we hypothesize that differences in habitat environments and diets may be potential drivers of significant differences in the gut microbial communities of domestic and wild mallards. In addition, the enrichment of potential pathogenic genera in the intestinal tract of domestic mallards should attract enough attention. It is noteworthy that we also lack the species- and strain-level assays needed to study disease and, in future work, we will rely on metagenomics technology to conduct more in-depth studies of bird gut microbial communities (including the function of gut microbes). This study provides basic information for the conservation of wild populations and the prevention and control of diseases in domestic mallards.

## Figures and Tables

**Figure 1 animals-13-02956-f001:**
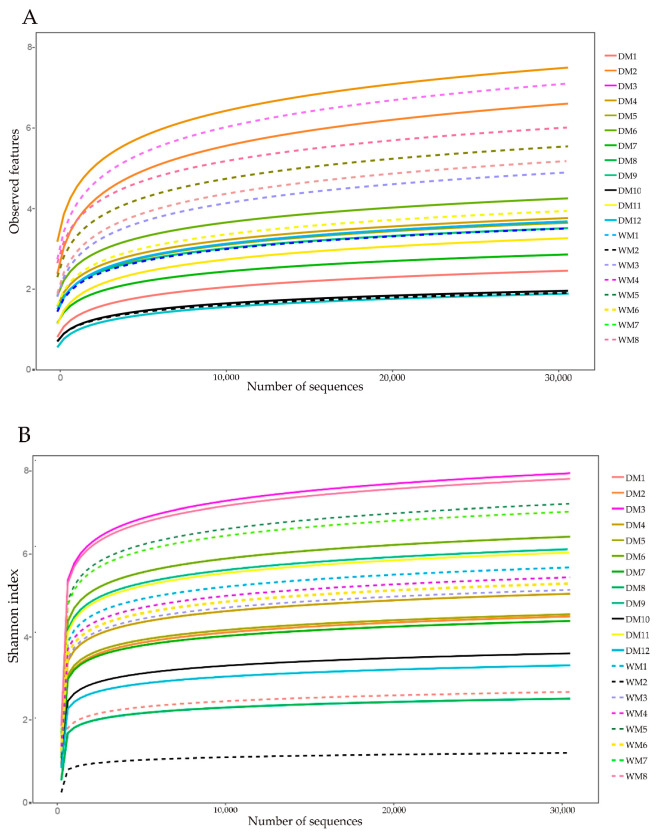
The rarefaction curves of observed features (**A**) and Shannon index (**B**) for the 20 samples. DM1-DM12 represent the samples collected in domestic mallards, and WM1–WM8 represent the samples collected in wild mallards.

**Figure 2 animals-13-02956-f002:**
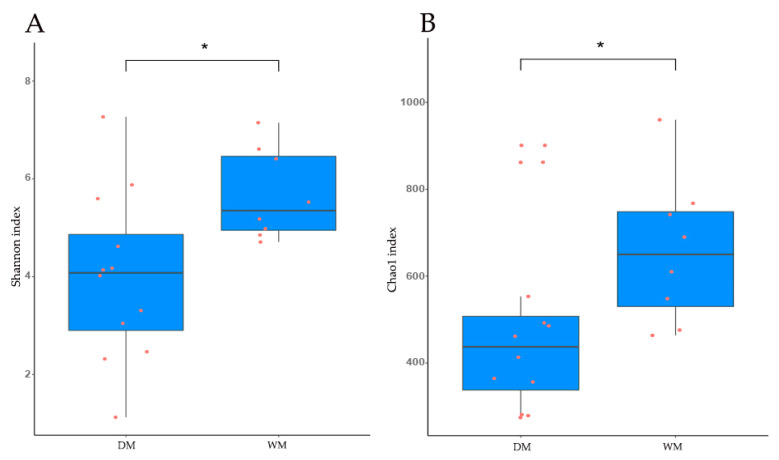
Alpha diversity indices of gut microbial communities in domestic mallards (DM) and wild mallards (WM). Shannon index (**A**) and Chao1 index (**B**) for DM and WM groups; * indicates *p* < 0.05.

**Figure 3 animals-13-02956-f003:**
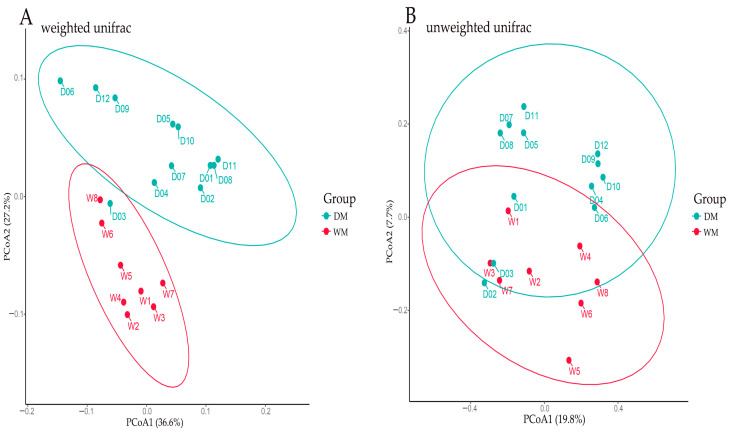
PCoA plot of samples using the weighted (**A**) and unweighted (**B**) UniFrac distance metric.

**Figure 4 animals-13-02956-f004:**
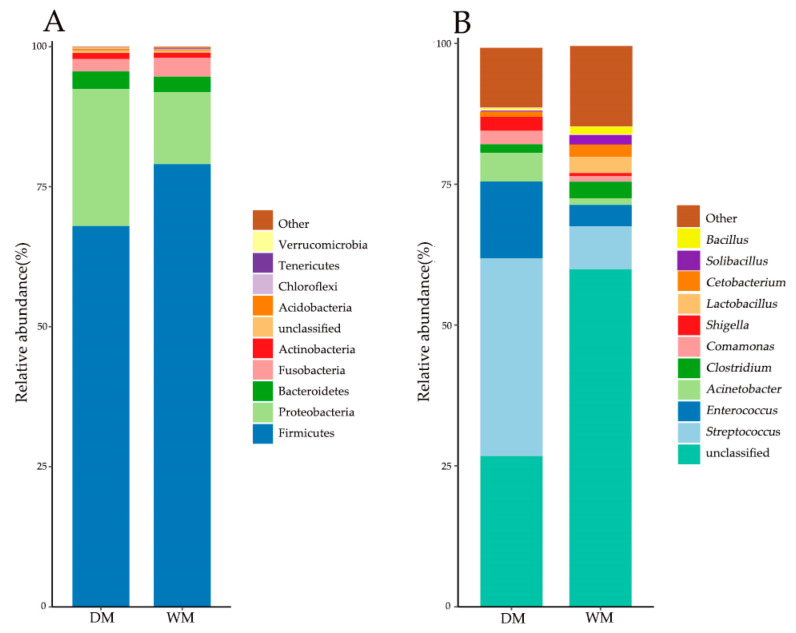
Top 10 phyla (**A**) and genera (**B**) in terms of mean relative abundance in the two sample groups.

**Figure 5 animals-13-02956-f005:**
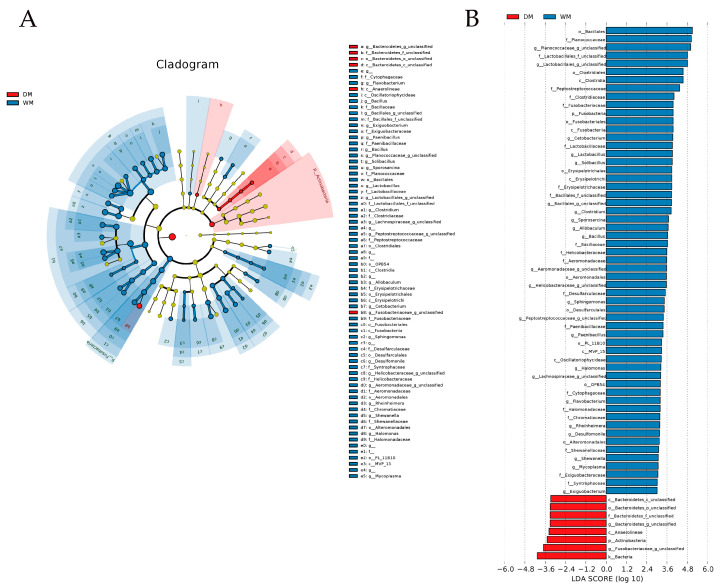
Linear discriminant analysis (LDA) effect size (LEfSe) analysis. (**A**) The cladogram diagram shows the microbial species with significant differences in the two groups. The cladogram diagram corresponds to the different taxonomic levels of the kingdoms, phyla, orders, families, and genera from the inside to the outside, and the lines between the levels represent the affiliation. Each circle node represents a species, and a yellow node means that the difference between groups is not significant, while a non-yellow node means that the species is a characteristic microorganism of the corresponding color group (with significantly higher abundance in that group). The colored sectors mark the subordinate taxonomic intervals of the characteristic microorganisms. (**B**) The plot from LEfSe analysis. The length of the bar column represents the LDA score. The figure exhibits the microbial community with significant differences between the two groups of mallards (LDA > 3.0, *p* < 0.05).

**Figure 6 animals-13-02956-f006:**
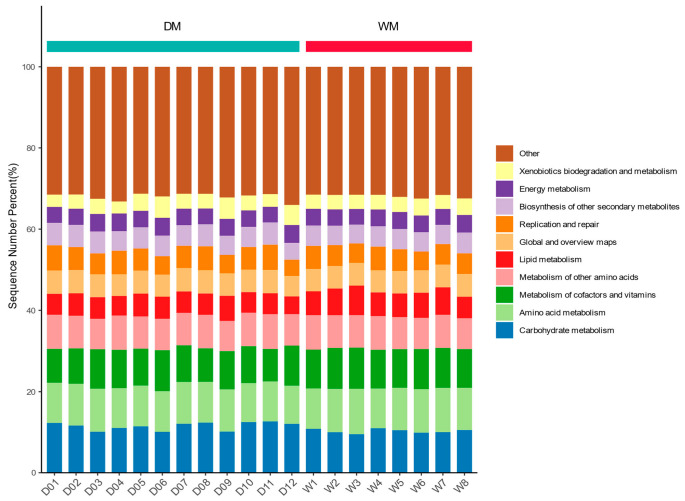
Top 10 level 2 pathways with predicted abundances of gut microbial function in the DM and WM groups.

**Table 1 animals-13-02956-t001:** Mean relative abundance ± SD > 1% of phyla in the two sample groups.

Category	Firmicutes	Proteobacteria	Bacteroidetes	Fusobacteria	Actinobacteria
WM	79.0% ± 10.2%	12.9% ± 9.5%	2.8% ± 2.4%	3.4% ± 2.5%	—
DM	68.0% ± 26.5%	24.5% ± 22.9%	3.1% ± 3.2%	2.2% ± 5.9%	1.1% ± 1.8%

**Table 2 animals-13-02956-t002:** Genera with a mean relative abundance ± SD > 1% in each phylum.

Phylum	Genus	DM	WM
Firmicutes	*Streptococcus*	35.1% ± 30.6%	7.6% ± 5.9%
	*Enterococcus*	13.7% ± 15.2%	3.8% ± 1.8%
	*Clostridium*	1.5% ± 1.2%	3.0% ± 1.4%
	*Lactobacillus*	—	2.8% ± 2.0%
	*Solibacillus*	—	1.9% ± 1.3%
	*Bacillus*	—	1.3% ± 0.9%
Proteobacteria	*Acinetobacter*	5.1% ± 6.9%	1.2% ± 1.1%
	*Comamonas*	2.4% ± 5.2%	—
	*Shigella*	2.5% ± 7.3%	—
Fusobacteria	*Cetobacterium*	—	2.2% ± 2.0%

## Data Availability

The raw data have been submitted to the NCBI Sequence Read Archive (BioProject ID: PRJNA995419).

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
