# Peer review of "Comparison of the Gut Microbial Communities of Domestic and Wild Mallards (Anas platyrhynchos) Based on High-Throughput Sequencing Technology"

_animals, 2023, doi:10.3390/ani13182956_

Round 1
Reviewer 1 Report
1. Lines 22-23: The main results of alpha diversity, eg Chao 1 and Shannon, is suggested to be specifc.
2. Lines 31-32: a specific genus is not appropriately expressed as a pathogen. Certain genus may contain several or many pathogens, and nonpathogens can be included in the same genus.
3. Lines 45-46: what about the link between gut microbiota and intestinal disorders/diseases? Typical examples of intestinal disorders/diseases linked with gut microbiota need to be added. In addition, dysbiosis should be discussed.
4. ‘Breeding environment’ seems to be related to potential genetic variation which can be a factor to cause difference in gut microbiota. Herein, rearing environment will be better if wild and domestic mallard have same genetic background. Otherwise, ‘breeding environment’ might cause confusion about their genetics.
5. What do we know about gut microbiota of wild or domestic mallard? In other word, is there any report on this? If yes, related studies should be reviewed in the introduction. If not, the novelty of this study should be addressed.
6. Linse 107-109 “Genomic DNA was extracted……, take an appropriate amount of sample in a centrifuge tube…….” Sentence should be restructured to correct the grammar.
7. Line 113: H2O
8. What are the P and R2 values of weighted and unweighted UniFrac distances in Figure 3, which need to be added in the plot?
9. Genus should be italicized in Table 2 and through text.
10. In figure 4, the title of y-axis is suggested to be “Relative Abundance”
11. Did you identify differential bacteria between the two groups? LEfSe can achieve this.
12. Why were fecal samples of wild and domestic mallard collected at different seasons? As you mentioned, season is one of factors that affect gut microbiota. Would season be a variation source of the results? This should be discussed in the discussion.
Grammar and language can be improved.
Author Response
Dear reviewer
First of all, I would like to thank you for your valuable comments on the content of my article, which have helped me a lot in revising my manuscript.You are very professional in all these comments you make and I am in admire of your responsible attitude.
Comments and Suggestions |
Reply |
1. Lines 22-23: The main results of alpha diversity, eg Chao 1 and Shannon, is suggested to be specifc. |
Line 23-26. |
2. Lines 31-32: a specific genus is not appropriately expressed as a pathogen. Certain genus may contain several or many pathogens, and nonpathogens can be included in the same genus. |
Line 401-410. |
3. Lines 45-46: what about the link between gut microbiota and intestinal disorders/diseases? Typical examples of intestinal disorders/diseases linked with gut microbiota need to be added. In addition, dysbiosis should be discussed. |
Line 48-52. |
4. ‘Breeding environment’ seems to be related to potential genetic variation which can be a factor to cause difference in gut microbiota. Herein, rearing environment will be better if wild and domestic mallard have same genetic background. Otherwise, ‘breeding environment’ might cause confusion about their genetics.
|
Changed to “habitat environment”。Line 14,88, 93,422,324. |
5. What do we know about gut microbiota of wild or domestic mallard? In other word, is there any report on this? If yes, related studies should be reviewed in the introduction. If not, the novelty of this study should be addressed. |
Line 99-110. |
6. Linse 107-109 “Genomic DNA was extracted……, take an appropriate amount of sample in a centrifuge tube…….” Sentence should be restructured to correct the grammar. |
Line 138. |
7. Line 113: H2O |
Line 145. |
8. What are the P and R2 values of weighted and unweighted UniFrac distances in Figure 3, which need to be added in the plot? |
Line 216-217. |
9. Genus should be italicized in Table 2 and through text. |
Line 274. |
10. In figure 4, the title of y-axis is suggested to be “Relative Abundance” |
Line 265. |
11. Did you identify differential bacteria between the two groups? LEfSe can achieve this. |
Line 254-262, Figure 5 |
12. Why were fecal samples of wild and domestic mallard collected at different seasons? As you mentioned, season is one of factors that affect gut microbiota. Would season be a variation source of the results? This should be discussed in the discussion. |
Line 132-136. |
Finally, once again, I would like to express my gratitude to you. Your help will help me go further in my research path. I wish you good health and success in your work.

Reviewer 2 Report
Review of Animals 10.3390 “Comparison of the gut microbial communities of domestic and wild mallards (Anas platyrhynchos) based on high-throughput sequencing technology”
The authors contrasted the microbiome from fecal samples of domestic and wild Mallard ducks.
Overall, the manuscript is well written and the Results and Discussion sections appear to be will supported by the data.
Since the sample was feces, the terms referring to guts, gastrointestinal tract, intestines, and stool should all be changed to feces. In the title “gut microbial” should be change to “fecal”.
The authors can continue to state that the fecal microbiome was used to infer or predict the composition of the intestinal or gut microbiome. However, the Authors should read “16S rRNA gene-based assessment of common broiler chicken sampling methods: Evaluating intra-flock sample size, cecal pair similarity, and cloacal swab similarity to other alimentary tract locations” in Front. Physiol. 13:996654. p.1-11. 2022. doi.org/10.3389/fphys.2022.996654
Similarly, the word “food” should be replaced with “diet” throughout the manuscript.
Specific comments:
L11 The authors state that “including gender” as a factor influencing gut bacteria, but I could not locate within the References a manuscript that supports this statement.
In Figure 1 for the rarefaction curves of observed (A) and Shannon index (B), with the colors chosen, this reviewer is not able to differentiate the domestic from the wild Mallards. Suggest representing the wild Mallards with dashed lines and the domestic Mallards with solid lines.

Author Response
Dear reviewer
First of all, I would like to thank you for your valuable comments on the content of my article, which have helped me a lot in revising my manuscript.You are very professional in all these comments you make and I am in admire of your responsible attitude.
Comments and Suggestions |
Reply |
(1)Since the sample was feces, the terms referring to guts, gastrointestinal tract, intestines, and stool should all be changed to feces. In the title “gut microbial” should be change to “fecal”. |
Since wild birds are prohibited from being killed indiscriminately, and after reviewing a large body of literature, we found that most bird researchers use fecal samples to represent the gut microbiota of birds for their studies. |
(2)The authors can continue to state that the fecal microbiome was used to infer or predict the composition of the intestinal or gut microbiome. However, the Authors should read “16S rRNA gene-based assessment of common broiler chicken sampling methods: Evaluating intra-flock sample size, cecal pair similarity, and cloacal swab similarity to other alimentary tract locations” in Front. Physiol. 13:996654. p.1-11. 2022. doi.org/10.3389/fphys.2022.996654
|
Line 115-121. |
(3)Similarly, the word “food” should be replaced with “diet” throughout the manuscript.
|
Line12,61,289,296,308,406. |
(4)L11 The authors state that “including gender” as a factor influencing gut bacteria, but I could not locate within the References a manuscript that supports this statement.
|
Line 62. |
(5)In Figure 1 for the rarefaction curves of observed (A) and Shannon index (B), with the colors chosen, this reviewer is not able to differentiate the domestic from the wild Mallards. Suggest representing the wild Mallards with dashed lines and the domestic Mallards with solid lines.
|
Figure 1
|
Finally, once again, I would like to express my gratitude to you. Your help will help me go further in my research path. I wish you good health and success in your work.

Reviewer 3 Report
In this paper, He et al. studied the differences in the gut microbial communities between domestic and wild mallards by 16S rRNA gene sequencing, and results showed that wild mallards had a higher diversity of microbiomes than domesticated mallards and a lower relative abundance of potential pathogens. The topic of this paper is pretty interesting. Here are some comments on this paper:
1. Line 18 “Mallard” should be “Mallards”, and line 24 “PCOA” should be “PCoA”. Lines 22-23 “Alpha diversity analysis..” I propose the authors could point out which alpha diversity was higher.
2. In the introduction, lines 45-47 I suggest the authors could the description of the role of gut microbiome in birds. Reference 8 was on the human diabetes.
3. The wild mallard fecal samples were collected from Caohai National Nature Reserve in March, while the domesticated mallards were collected from June to August 2022 at Guangxi University. Could authors provide reasons why fecal samples were collected in different areas and at different times? The fecal samples of the domesticated mallards should also be collected from Caohai National Nature Reserve in March. As the authors described in lines 258-260 “Birds in different locations feed on different foods, and microorganisms ingested with the food may be one of the main pathways for microbial colonization of the gastrointestinal tract of birds”.
4. Missing statistical analysis information in the methods section.
5. Lines 204-211, I was quite confused by the results of the LEsfe analysis. The LEsfe analysis method is supposed to be used to identify different taxa between wild and domesticated mallards.
6. The results of relative abundance shall be presented as mean ± sd.
7. Line 242 “Cancer: overview”, could authors provide any explanation?
8. Some formatting issues in the paper, like line 113 “H2O”, line 131 “dada2”, and line 171 “PCOA”.
Author Response
Dear Reviewer,
First of all, I would like to express my gratitude to you for your timely letter. Secondly, I would like to thank the reviewers for their comments on the content of my article.The comments made by the reviewers were not only very professional, but their attitude of responsibility was very admirable to me. Below, I will respond to each of the comments made by the reviewers.
(1) PCOA changed to PCoA in article.
(2) In all the current studies there is no information on the exact role of gut microorganisms in wild birds, and we can only refer to information on studies of gut microorganisms in humans and other animals. The main purpose is to highlight the importance and physiological significance of studying gut microbes。
(3) The main purpose of this article is to explore whether there are significant differences in the fecal microbial communities of domestic and wild mallard ducks under different habitat conditions of the same species. It does not involve exploring the effect of seasonal factors on fecal microbes for the time beingThe seasonal factors mentioned in the article refer to research by others. It also provides ideas for studying fecal microbes in wild birds. I should take note of this in my future research.
(4) The results of LEfSe analysis showed no significant differences in biomarkers at the phylum and genus level in domestic mallards. Therefore the results of LEfSe analysis only showed biomarkers that were significantly different in the wild mallard group.
Finally, once again, thank you for your hard work. I wish you good health and success in your work, thank you.

Round 2
Reviewer 3 Report
I appreciate for authors’ response and modifications to the paper. After carefully reviewing the authors’ response and the revised manuscript. I found that the authors did not respond to my concerns point by point, particularly comments 4 and 6 regarding “statistical analysis” and “forms of relative abundance”.
1. Missing statistical analysis information in the methods section. Without statistical information, we don't know how to do the difference analysis, such as in Figure 2 “* indicates p < 0.05”.
2. Relative abundance results should be expressed as mean ± sd, such as Tables 1 and 2.
3. The page 9 is blank.
4. I was still confused by the way the LEsfe analysis was described. As the authors mentioned in lines 168-169 “LEfSe was employed to identify bacteria that differed in abundance between groups and samples”. Therefore, compared to domestic mallards, wild mallards showed significant differences in Deferribacteres (p < 0.05), the Tenericutes (p < 0.001), and the Fusobacteria (p < 0.01).
5. “The main purpose of this article is to explore whether there are significant differences in the fecal microbial communities of domestic and wild mallard ducks under different habitat conditions of the same species. It does not involve exploring the effect of seasonal factors on fecal microbes for the time beingThe seasonal factors mentioned in the article refer to research by others. It also provides ideas for studying fecal microbes in wild birds. I should take note of this in my future research.”
Reply: Since it has been suggested in the literature that season affects the gut microbiome of birds. The authors should have taken this factor into account when designing and conducting the experiment. This is an experimental design flaw.
Author Response
Dear reviewers,
These comments you have made are very timely and professional and have helped me a lot in revising my article. Once again, I believe the hard work is appreciated
Question |
Response |
1. Missing statistical analysis information in the methods section. Without statistical information, we don't know how to do the difference analysis, such as in Figure 2 “* indicates p < 0.05”.
|
Line 177-185 |
2. Relative abundance results should be expressed as mean ± sd, such as Tables 1 and 2.
|
Line28-32, 234-238, 265, 273. |
3. I was still confused by the way the LEsfe analysis was described. As the authors mentioned in lines 168-169 “LEfSe was employed to identify bacteria that differed in abundance between groups and samples”. Therefore, compared to domestic mallards, wild mallards showed significant differences in Deferribacteres (p < 0.05), the Tenericutes (p < 0.001), and the Fusobacteria (p < 0.01).
|
Figure 5 |
4. “The main purpose of this article is to explore whether there are significant differences in the fecal microbial communities of domestic and wild mallard ducks under different habitat conditions of the same species. It does not involve exploring the effect of seasonal factors on fecal microbes for the time being.The seasonal factors mentioned in the article refer to research by others. It also provides ideas for studying fecal microbes in wild birds. I should take note of this in my future research.”
|
Line 132-136 |

Round 3
Reviewer 3 Report
Thanks to the authors for replying. In addition to the figures' clarity needing to be improved, I have no further concerns.